# **Dual-lidar profilers for measuring atmospheric turbulence**

Maxime Thiébaut and Neil Luxcey

France Énergies Marines, Technopôle Brest-Iroise, 525 Avenue Alexis de Rochon, 29280 Plouzané, France

**Correspondence:** Maxime Thiébaut (maxime.thiebaut@france-energies-marines.org)

**Abstract.** A dual-lidar system, consisting of WindCube v2.1 profilers with one lidar oriented horizontally at 45° relative to the other, was deployed to estimate along- and cross-wind variances, velocity spectra, and turbulence intensity (TI) using the "variance method". This approach computes second-order statistics directly from the line-of-sight (LOS) velocity variances to infer the full three-dimensional velocity variance components. It is benchmarked against the "traditional method" commonly used in the wind energy sector, which reconstructs instantaneous velocity components from single-lidar LOS measurements before deriving second-order statistics. Both methods are evaluated at a single altitude using a 30-day collocated dataset, with reference measurements from a sonic anemometer classified by atmospheric stability conditions. Two key performance metrics are considered: the mean relative bias error (MRBE) and the relative root mean square error (RRMSE), as defined by DNV (Det Norske Veritas). Spectral analysis of the velocity components shows that, while the traditional method more closely matches the reference spectra at low frequencies, it tends to overestimate spectral energy at higher frequencies. In contrast, the variance method typically underestimates spectral energy in the along-wind component and overestimates it in the crosswind component. Furthermore, linear regression analysis reveals that the variance method captures 90-97% of the reference variances across all stability regimes, while the traditional method tends to overestimate—especially for cross-wind variance under unstable and neutral conditions (up to 132%). Overall, the variance method yields lower MRBE and RRMSE values for both along- and cross-wind TI. Specifically, for along-wind TI, MRBE decreases from 10.1% to 7.5% and RRMSE from 19.5% to 11.7%. For cross-wind TI, MRBE is reduced from up to 17.6% to 12.3%, and RRMSE from 23.4% to 18.2%.

### 1 Introduction

In recent years, wind lidar profiler technology has increasingly replaced traditional meteorological masts equipped with insitu sensors such as cup or sonic anemometers for measuring key mean wind properties like speed and direction. Wind lidars can be broadly categorized based on their emission waveform, pulsed or continuous, and their associated measurement techniques. Typically, pulsed lidar profilers employ the Doppler Beam Swinging (DBS) technique (Strauch et al., 1984), where the instrument samples line-of-sight (LOS) velocities sequentially along several fixed beam directions, while continuous-wave lidars use the Velocity-Azimuth Display (VAD) method (Browning and Wexler, 1968), where LOS velocities are measured around a conical scan at a fixed elevation angle (e.g., Mann et al., 2010).

Lidar profilers offer clear advantages for mean wind properties characterization, including reduced deployment costs and the ability to measure at similar or greater heights above ground compared to meteorological masts. However, wind lidar profilers

have yet to garner widespread acceptance for turbulence measurement, which remains a focal point of ongoing research. In contrast to turbulence data derived from reference instruments such as cup or sonic anemometers, turbulence estimates from lidar profilers are subject to systematic errors resulting from three primary sources: (i) inter-beam effect, (ii) intra-beam averaging, and (iii) instrumental noise. The inter-beam effect can lead to over- or underestimation of turbulence metrics due to modulation of energy from eddies of specific wavenumbers (Theriault, 1986; Gargett et al., 2009; Kelberlau and Mann, 2020). This issue is tightly linked to the assumption of instantaneous spatial homogeneity in multi-beam techniques; when violated, it introduces inter-beam variance contamination. Intra-beam effect stems from probe-time averaging and introduces an underestimation of turbulent energy. This bias results from two types of anisotropic filtering: (1) spatial filtering across the lidar probe volume and (2) temporal filtering due to pulse accumulation time (Thiébaut et al., 2025).

Instrumental noise contributes to turbulence overestimation by adding variance to the measured LOS velocities. For the WindCube v2.1, Thiébaut et al. (2025) found this effect to be about 5% of the total variance. Noise can be identified and removed via spectral analysis, where it appears as a flat high-frequency component (e.g., Hurther and Lemmin, 2001; Durgesh et al., 2014), or estimated through indirect corrections. In the latter, noise variance is modeled as a function of sample volume distance from the sensors (Lhermitte and Lemmin, 1994; Zedel et al., 2002), inferred from two-point cross-correlation assuming uncorrelated noise between locations (Garbini et al., 1982), or derived from the zero-lag offset in the auto-correlation function, exploiting the lack of temporal correlation in noise (e.g., Lenschow et al., 2000).

Turbulence intensity (TI) is a key metric in wind energy for evaluating turbine loads, site viability, and predicting energy yield. The traditional approach computes TI from second-order statistics derived from instantaneous reconstructed velocity components using LOS measurements from a single-lidar profiler. However, this method combines the effects of inter- and intra-beam effects and instrumental noise. In some cases, these biases may cancel each other out, resulting in correct TI estimates for the wrong reasons (Kelberlau and Mann, 2019). An alternative is to derive TI from second-order statistics of the three-dimensional velocity components, computed directly from the variances of the LOS velocities (Eberhard et al., 1989). Applied to a 5-beam pulsed wind lidar profiler, this method allows the estimation of five components of the Reynolds stress tensor in the instrument coordinate system. In this system, the x-axis is aligned with one pair of opposing slanted beams, and the y-axis with the other. The only component that cannot be resolved is the cross-term variance  $\sigma_{xy}$ . However, this term is essential for rotating the variances in the x and y directions to obtain the streamwise variance. In the absence of  $\sigma_{xy}$ , the method can be applied only when the wind direction aligns with a pair of opposite slanted beams (e.g., Thiébaut et al., 2024).

To measure the Reynolds stress component  $\sigma_{xy}$ , an additional slanted beam is required. Sathe et al. (2015) proposed such a configuration using a ground-based scanning pulsed lidar system called WindScanner. This device enables the computation of LOS velocity variances at five equally spaced azimuth angles on the base of a scanning cone, along with one additional measurement at the center of the scanning circle using a vertically oriented beam at the same height. The authors estimated TI by deriving second-order statistics of the three-dimensional velocity components from the variances of the LOS velocities. They compared these estimates to those obtained using a continuous-wave lidar profiler based on the VAD technique and to measurements from a reference cup anemometer. The results show that, depending on atmospheric stability and the wind field component, the 6-beam method captures between 85% and 101% of the reference turbulence, whereas the VAD-based method

70

captures between 66% and 87%. This indicates a significant improvement in turbulence characterization using the 6-beam approach. It is important to note that the WindScanner is not a commercial device but rather a scientific instrument used by a limited number of research groups, which constrains its broader application in operational settings. Moreover, the duration of a complete 6-beam measurement cycle with the WindScanner was approximately 15 s, corresponding to a LOS velocity sampling rate of 0.067 Hz. Applying Taylor's frozen turbulence hypothesis, which states that eddies of size L and frequency f are advected by the mean flow velocity U, such that  $f \approx U/L$ , this sampling rate implies that, for a mean wind speed of 10 m/s, only eddies larger than approximately  $L \approx 150$  m can be resolved. For optimized wind turbine design, however, it is essential to capture smaller turbulence structures and their associated energy content.

In oceanographic studies, Acoustic Doppler Current Profilers (ADCPs) are used to derive first- and second-order statistics of water flow. An ADCP is a Doppler-based remote sensing instrument that uses multiple beams to measure flow velocity. A common configuration is the Janus setup, as in the Workhorse Sentinel by Teledyne RDI, which features four beams inclined  $20^{\circ}$  from the vertical. The WindCube v1 lidar employed a similar multi-beam approach but with a steeper beam inclination. Other ADCPs, such as the Nortek Signature series, use a 5-beam Janus configuration with four beams slanted at  $25^{\circ}$  and one vertical beam—closely resembling the current WindCube v2.1, which combines four slanted beams at  $\alpha = 28^{\circ}$  from the vertical with a vertical beam. ADCPs can be deployed bottom-mounted in an upward-looking configuration (e.g., Thomson et al., 2012; McMillan et al., 2016; Thiébaut et al., 2022) or in a downward-looking configuration when installed on a floating structure or towed by vessels (e.g., Goddijn-Murphy et al., 2013; Sentchev et al., 2019; Thiébaut et al., 2019).

The method that uses second-order statistics of LOS velocity to reconstruct the variance of three-dimensional velocity components can also be applied to ADCP measurements. In ocean sciences, this approach is commonly referred to as the variance method (e.g., Stacey et al., 1999a; Lu and Lueck, 1999; Rippeth et al., 2002). Turbulence analysis using a single-ADCP (up to five beams) typically relies on assumed turbulence anisotropy ratios, often derived from laboratory studies (e.g., Stacey et al., 1999b; Lueck et al., 2002; Peters and Johns, 2006). However, Burchard et al. (2008) argue that this approach is problematic, as the difference between fully anisotropic and isotropic turbulence can lead to a six-fold discrepancy in estimated turbulent kinetic energy (TKE) using a Teledyne RDI ADCP. These assumptions are necessary because it is not possible to resolve all six independent components of the Reynolds stress tensor with fewer than six beams.

To overcome this limitation, Vermeulen et al. (2011) proposed a dual-ADCP setup using two 4-beam RDI instruments with slanted beams. The second ADCP was oriented horizontally with a 45° heading offset relative to the first, forming an 8-beam configuration. This specific orientation was shown numerically to minimize error in velocity variance estimates. Additionally, the second ADCP was pitched at 20° to align one beam vertically upward, enabling direct measurement of the vertical velocity component. With eight independent beams, assumptions about anisotropy become unnecessary, and the full Reynolds stress tensor can be resolved and rotated into any coordinate frame. This configuration was later implemented by Thiébaut et al. (2020) to investigate the TKE budget in the highly energetic Alderney Race (France). Mercier et al. (2021) further validated the method by replicating the 8-beam setup in large-eddy simulations.

Building on the methodology introduced by Vermeulen et al. (2011) for ADCPs, we developed a modified version tailored to dual-WindCube v2.1 lidar profilers. The dual-lidar system is used to estimate along- and cross-wind variances, velocity

spectra, and TI using the variance method. This method is benchmarked against the traditional method used in the wind power industry. Both methods are evaluated at a single altitude using a 30-day collocated dataset, with reference measurements from a 3D sonic anemometer classified by atmospheric stability conditions. Two key performance metrics are considered: the mean relative bias error (MRBE) and the relative root mean square error (RRMSE), as defined by DNV (Det Norske Veritas) (DNV, 2023).

The remainder of this paper is organized as follows. The study begins with a description of the study site (Sect. 2.1), instrumentation (Sect. 2.2), and data selection (Sect. 2.3), along with methods for instrumental noise correction and spike filtering (Sects. 2.4 and 2.5). This section also presents the traditional method for single-lidar profilers, the variance method for dual-lidar profilers, and the reference sonic anemometer measurements used to derive velocity variances, TI (Sect. 2.6), and velocity power spectral density (Sect. 2.7). Atmospheric stability classification based on the sonic anemometer measurements is described in Sect. 2.8. Error metrics for method evaluation are defined in Sect. 2.9. The results of the comparative analysis between the variance and traditional methods are presented in Sect. 3, followed by a discussion in Sect. 4. Finally, Sect. 5 summarizes the main findings and provides concluding remarks.

#### 110 2 Data and methods

# 2.1 Study site and meteorological mast

The measurement mast has been installed since February 2015, located 13 km offshore from Fécamp along the coast of Normandy, France (Fig. 1). Initially operated by EDF (Électricité de France), the operation of the mast was transferred to France Énergies Marines at the end of 2022. The entire structure, including the foundation and measurement equipment, stands 90 m tall. The lattice tower, which supports various sensors, reaches a height of 40 m, while the top of the mast rises approximately 60 m above sea level, depending on tidal variations. The average tidal range at the site is 8.8 m. The mast is anchored to a reinforced concrete structure, resembling gravity-based foundations, with a diameter of 23 m and a total weight of 1,800 tons, placed directly on the seabed.

The mast was initially constructed to characterize the site for the wind farm deployment, providing essential data for the planning and design phases, and to test the foundation, which could potentially be used for the wind turbine foundations. It is positioned in front of the first row of wind turbines to the south of the wind farm, with the nearest turbine located 400 m to the west. The 497-megawatt (MW) wind project, which began operating in May 2024, comprises 71 Siemens Gamesa Renewable Energy 7-MW turbines. These turbines are located between 13 and 24 km off the coast of Fécamp, at depths of approximately 30 m (Fig. 1b). The turbines span an area of about 60 km<sup>2</sup>.

# 125 2.2 Sensors equipment

The measurement mast is equipped with a 3D THIES sonic anemometer mounted on a boom at a height of 39 m above the platform (Fig. 2a). The measurement volume of the sonic anemometer is positionned at 39.5 m, as it is located 0.5 m from the

135

140

**Figure 1.** Photograph of the measurement mast (a), located 13 km off the coast of Fécamp, situated in front of the first row of wind turbines in the offshore wind farm (b), which is deployed off the coast of Normandy, France (c).

tip of the boom. The boom is oriented at an angle of -60° from True North (Fig. 2b). The anemometer is aligned such that the x-axis points toward True North, the y-axis toward the West, and the z-axis vertically upward. The (x, y, z) coordinate system associated with the sonic anemometer served as the reference coordinate system in this study. The anemometer collects wind velocity data at an acquisition rate of 10 Hz.

To complement this setup, two Vaisala offshore WindCube v2.1 lidar profilers were installed at the platform level (Fig. 2b and Fig. 3bc). The primary lidar, referred to as lidar A, was oriented such that beams 1 and 3—aligned with the x-axis of the instrument's coordinate system—were directed toward True North, while beams 2 and 4 were aligned with the y-axis. The fifth beam of lidar A pointed vertically along the z-axis. Note that the manufacturer configured the z-axis of the lidar to point vertically downward. The secondary lidar, referred to as lidar B, was mounted adjacent to lidar A and rotated at  $\theta = 45^{\circ}$  in the horizontal plane relative to lidar A. Both lidars were installed with a tilt angle of less than  $0.2^{\circ}$ . Together, the two lidars formed a combined 10-beam scanning arrangement (Fig. 3a), enabling measurements of LOS velocities at ten vertical levels ranging from 40 m to 240 m above the platform. LOS velocity data were recorded at a sampling rate of 0.25 Hz. To enable a consistent comparison, the sonic anemometer dataset was downsampled to the same rate. This step is critical for ensuring that all measurements are temporally aligned and comparable. The downsampling was performed using a decimation approach, which reduces the number of data points by an integer factor while applying a low-pass filter. This effectively reproduces what a device operating at the lower sampling rate would measure, capturing the energy of larger eddies but necessarily filtering out smaller, high-frequency turbulent fluctuations that cannot be resolved at the lower rate.

Additionally, atmospheric data are collected by a Vaisala WXT 530 weather transmitter, installed on the mast at a height of 39 m above the platform. This sensor provides 10-min averaged values of pressure, temperature, and humidity.

#### 2.3 Data selection

A 30-day dataset spanning from April 1 to May 1, 2024, was compiled for turbulence analysis. This dataset was divided into 1,440 non-overlapping 30-min ensembles. The choice of a 30-min averaging window—rather than the traditional 10-min

**Figure 2.** Side view (a) and top view (b) schematics of the 40-m measurement mast, detailing the deployed sensors and their positions on the mast and platform.

**Figure 3.** (a) Schematic of the 10-beam arrangement formed by combining the five beams of lidar A with the five beams of lidar B, which is rotated 45° relative to lidar A. (b) Side view and (c) top view photographs of the setup installed at the platform level.

interval commonly used in the wind energy industry—was motivated by the goal of reducing random errors in turbulence measurements, in accordance with the recommendations of Lenschow et al. (1994). The analysis focused on wind speeds exceeding 3 m/s, corresponding to the typical cut-in speed of most wind turbines. Applying this threshold led to the exclusion of approximately 6% of the original dataset. Additionally, wind measurements associated with wind directions between 120° and 180° (relative to True North) were excluded to avoid contamination from mast wake effects affecting the sonic anemometer data. This directional filtering removed a further 11% of the remaining ensembles. To ensure data quality, any 30-min ensemble containing less than 90% valid velocity measurements was also discarded, resulting in an additional 9% reduction. Individual velocity measurements were considered valid only if their carrier-to-noise ratio (CNR), provided by the lidar systems, was greater than -23 dB, as recommended by the manufacturer. After applying all filtering criteria, a total of N = 1,098 30-min ensembles were retained for turbulence analysis. The final dataset spans wind speeds ranging from 3 to 21 m/s, with a mean wind speed of 9.7 m/s and a median of 9.3 m/s.

# 2.4 Instrumental noise

Lidar measurements are inherently influenced by signal noise and potential variations in aerosol fall speeds, both of which contribute additional terms to the observed variance. Assuming that all atmospheric flow contributions to the observed LOS

velocity variance within the considered short timescales are of a turbulent nature, the variance  $\sigma_{b_i}^2$  of the LOS velocity measured by beam i, can be expressed as the sum of three independent terms (Doviak and Zrnic, 1993):

$$\sigma_{b_i}^2 = \sigma_{p_i}^2 + \sigma_{n_i}^2 + \sigma_{d_i}^2 \tag{1}$$

Here,  $\sigma_{p_i}^2$  represents the net contribution from atmospheric turbulence at scales measurable by the lidar (Brugger et al., 2016),  $\sigma_{n_i}^2$  denotes the variance associated with instrumental noise, and  $\sigma_{d_i}^2$  accounts for the variance caused by variations in aerosol terminal fall speeds within the probe volume. However,  $\sigma_{d_i}^2$  can typically be neglected, as particle fall speeds are generally less than 1 cm/s (e.g., Bodini et al., 2018).

The noise contribution  $\sigma_{n_i}^2$  is often quantified through an autocorrelation approach. In this method, the temporal autocorrelation function of the measured LOS velocity time series,

$$R(\tau) = \frac{\langle b_i(t) \, b_i(t+\tau) \rangle}{\sigma_{b_i}^2},\tag{2}$$

is examined at short time lags  $\tau$ . While atmospheric fluctuations remain correlated over small  $\tau$ , the noise component is 175 uncorrelated and manifests as a discontinuity at the first nonzero lag. The reduction of  $R(\tau)$  between  $\tau = 0$  and  $\tau = \Delta t$ can therefore be used to estimate  $\sigma_{n_i}^2$  (e.g., Lenschow et al., 2000). For a wind lidar profiler such as the WindCube v2.1, the sampling interval  $\Delta t$  corresponds to the accumulation time of the Doppler signal at one LOS position. This separation enables the retrieval of the turbulence-related variance  $\sigma_{p_i}^2$  with reduced contamination from instrumental effects. Note that, for consistency, instrumental noise were similarly removed from variance of the time series of the instantaneous reconstructed horizontal velocities,  $u_x$  and  $u_y$ , employed in the traditional method (Sect. 2.6.2).

#### Spike filtering 2.5

To remove spurious outliers in the velocity measurements, a spike filtering procedure following Wang et al. (2015) was applied. This method is designed to identify unrealistic spikes in wind signal records that may arise from intermittent low signal-to-noise ratios or other measurement artifacts. The filter operates on the time series by evaluating the deviation of each data point from the local median within a moving window. If the deviation exceeds a specified threshold, typically defined in terms of multiples of the local standard deviation, the point is flagged as a spike and subsequently removed or replaced through interpolation. In the present study, this filtering technique was applied consistently across all datasets, including the LOS velocity time series from the lidar profilers, the reconstructed horizontal wind components  $u_x$  and  $u_y$  from the lidar, and the three-dimensional velocity components measured by the sonic anemometer. This ensures that spurious outliers are removed from both remote sensing and in situ measurements, thereby improving the robustness of the turbulence statistics derived from these observations.

#### **Reconstruction of turbulence metrics** 2.6

#### Sonic anemometer 2.6.1

To reduce the influence of alignment and tilt errors on the variance estimates, a two-step coordinate rotation is applied to the sonic anemometer data (Foken and Mauder, 2008). Following the procedure described by Kaimal and Finnigan (1994), the

coordinate system of the sonic anemometer (x, y, z) is first rotated such that the mean velocity component  $\langle u_y \rangle$ —where the brackets denote a temporal average over a 30-min period—becomes zero, and the  $u_x$  component aligns with the mean wind direction  $\Theta$  defined as:

$$\Theta = \arctan\left(\frac{\langle u_y \rangle}{\langle u_x \rangle}\right) \tag{3}$$

This rotation yields a new intermediate coordinate system  $(x_1, y_1, z_1)$ , with transformed velocity components given by:

$$200 \quad u_{x_1} = u_x \cos\Theta + u_y \sin\Theta \tag{4}$$

$$u_{y_1} = -u_x \sin\Theta + u_y \cos\Theta \tag{5}$$

$$u_{z_1} = u_z \tag{6}$$

In the second rotation, the intermediate coordinate system  $(x_1, y_1, z_1)$  is further rotated in the vertical plane by an angle:

$$\phi = \arctan\left(\frac{\langle u_{z_1} \rangle}{\langle u_{x_1} \rangle}\right) \tag{7}$$

resulting in a final coordinate system  $(x_2, y_2, z_2)$ , in which the mean vertical velocity component is zero. The velocity components in this new system are given by:

$$u_{x_2} = u_{x_1}\cos\phi + u_{z_1}\sin\phi \tag{8}$$

$$u_{y_2} = u_{y_1} \tag{9}$$

$$u_{z_2} = -u_{x_1} \sin \phi + u_{z_1} \cos \phi \tag{10}$$

In this final rotated frame, the velocity components are denoted as  $u = u_{x_2}$ ,  $v = u_{y_2}$  and  $w = u_{z_2}$ , corresponding to the alongwind, cross-wind and vertical-wind components, respectively.

These components are then used to compute the reference along and cross-wind variances— $\sigma_u^2$  and  $\sigma_v^2$ —employed to derive the reference along and cross-wind turbulence intensity— $TI_u$  and  $TI_v$ —defined as:

$$TI_u = \frac{\sqrt{\sigma_u^2}}{U} \tag{11}$$

215

$$TI_v = \frac{\sqrt{\sigma_v^2}}{U} \tag{12}$$

where  $U=\sqrt{\langle u\rangle^2+\langle v\rangle^2}$  is the mean wind speed.

# 2.6.2 Single-lidar profiler — Traditional method

The traditional method for deriving turbulence information from lidar profiler measurements involves computing second-order statistics from reconstructed instantaneous velocity components, which are inferred from LOS velocities measured by a single-lidar profiler. Similarly to the sonic anemometer dataset, the reconstructed instantaneous horizontal velocities  $u_x$  and  $u_y$  from

the lidar were rotated in the horizontal plane to obtain  $(u_{x_1}, u_{y_1})$  using Eq. 4 and 5. A second rotation was not required due to the small tilt angle of the lidars. Therefore, after the first rotation, the lidar velocities can be expressed as  $u = u_{x_1}$  and  $v = u_{y_1}$ . These components are then used to compute the variances  $\tilde{\sigma}_u^2$  and  $\tilde{\sigma}_v^2$ . The tilde notation is used here to denote turbulence metrics derived using the traditional method. The along and cross-wind turbulence intensity— $\tilde{\mathrm{TI}}_u$  and  $\tilde{\mathrm{TI}}_v$ —derived from this method are then defined as:

$$\tilde{\mathrm{TI}}_u = \frac{\sqrt{\tilde{\sigma}_u^2}}{U} \tag{13}$$

$$\tilde{\mathrm{TI}}_v = \frac{\sqrt{\tilde{\sigma}_v^2}}{U} \tag{14}$$

# 230 2.6.3 Dual-lidar profilers — Variance method

The variance method involves the computation of second-order statistics of the three-dimensional velocity components from the LOS velocity variances. Building on the methodology introduced by Vermeulen et al. (2011) for ADCPs, we developed a modified version tailored to dual-WindCube v2.1 lidar profilers, each with a 5-beam configuration. Our approach constructs a ten-element vector  $\boldsymbol{b}$  containing noise-corrected LOS variances,  $\sigma_{p_i}^2$ , from both devices, allowing reconstruction of the full Reynolds stress tensor. The vector is defined as:

$$\boldsymbol{b} = \left[\sigma_{p_1,A}^2, \sigma_{p_2,A}^2, \sigma_{p_3,A}^2, \sigma_{p_4,A}^2, \sigma_{p_5,A}^2, \sigma_{p_1,B}^2, \sigma_{p_2,B}^2, \sigma_{p_3,B}^2, \sigma_{p_4,B}^2, \sigma_{p_5,B}^2\right]$$
(15)

where subscripts  $p_1$  to  $p_5$  denote the five individual beam of each lidar profiler, and the superscripts A and B refer to the primary and secondary lidar profilers, respectively. In Eq. 15,  $\sigma_{p_i,A}^2$  and  $\sigma_{p_i,B}^2$  are given by:

$$\sigma_{p_i,A}^2 = \int_0^\infty S_{i,A}(f)df \tag{16}$$

240

235

$$\sigma_{p_i,B}^2 = \int_0^\infty S_{i,B}(f)df \tag{17}$$

where  $S_{i,A}$  and  $S_{i,B}$  are the LOS velocity power spectral density measured by beam i of lidars A and B.

A transformation matrix T is then defined to project the LOS variances into the reference coordinate system (x, y, z). This matrix consists of two components: one associated with the beam geometry of the primary lidar, and the other corresponding to the secondary lidar. The latter includes a counterclockwise rotation matrix to account for the relative yaw angle between the two devices. The full transformation matrix is expressed as:

To extract the Reynolds stress tensor components from the LOS variances, an intermediate  $10 \times 6$  matrix  $\mathbf{Q}$  is computed, with elements defined as:

$$\mathbf{250} \quad \mathbf{Q} = T_{p,q} \cdot T_{p,m} \tag{19}$$

where p = 1, ..., 10 corresponds to the ten beams in the dual-lidar configuration, with beams 1 to 5 (denoted 1A, 2A, 3A, 4A and 5A in Fig. 3a) from the primary lidar and beams 6 to 10 (denoted 1B, 2B, 3B, 4B and 5B in Fig. 3a) from the secondary lidar, and q, m = 1, 2, 3 correspond to the three spatial directions in the reference coordinates system. The matrix,  $\mathbf{Q}$ , relates the LOS variance vector  $\mathbf{b}$  to the Reynolds stress vector  $\mathbf{r}$  as:

$$b = \mathbf{Q}r \tag{20}$$

Eq. 20 is overdetermined and is solved in a least-squares sense using the Moore–Penrose pseudoinverse:

$$r = (\mathbf{Q}^{\top} \mathbf{Q})^{-1} \mathbf{Q}^{\top} b = \mathbf{Q}^{+} b \tag{21}$$

Accuracy can be optimized by maximizing the determinant of  $\mathbf{Q}^{\top}\mathbf{Q}$ ; if this determinant approaches zero, the solution becomes ill-conditioned. In our case, since each lidar provides a spatially uniform distribution of beams, the only design variable that influences the determinant is the relative orientation between sensors. By offsetting lidar B by 45°, we maximize the angular spread of the combined sensing directions, thereby maximizing  $\det(\mathbf{Q}^{\top}\mathbf{Q})$  and improving the conditioning of the solution.

The six-element vector r contains the independent components of the Reynolds stress tensor,  $\mathbf{R}$ , which is reconstructed as a symmetric  $3 \times 3$  matrix:

265 
$$\mathbf{R} = \begin{pmatrix} r_1 & r_4 & r_5 \\ r_4 & r_2 & r_6 \\ r_5 & r_6 & r_3 \end{pmatrix} = \begin{pmatrix} \hat{\sigma}_x^2 & \hat{\sigma}_{xy} & \hat{\sigma}_{xz} \\ \hat{\sigma}_{xy} & \hat{\sigma}_y^2 & \hat{\sigma}_{yz} \\ \hat{\sigma}_{xz} & \hat{\sigma}_{yz} & \hat{\sigma}_z^2 \end{pmatrix}$$
 (22)

The hat notation is used here to denote turbulence metrics derived using the variance method.

The Reynolds stress tensor is subjected to the same initial coordinate rotation described in Sect. 2.6.1, resulting in transformed components within the rotated horizontal coordinate system  $(x_1,y_1)$ . In this frame, the variances of the along-wind and crosswind velocity components—denoted  $\hat{\sigma}_u^2 = \hat{\sigma}_{x_1}^2$  and  $\hat{\sigma}_v^2 = \hat{\sigma}_{y_1}^2$ , respectively—are given by:

$$270 \quad \hat{\sigma}_{u}^{2} = \hat{\sigma}_{x}^{2} \cos^{2} \Theta + \hat{\sigma}_{y}^{2} \sin^{2} \Theta + \hat{\sigma}_{xy} \sin(2\Theta) \tag{23}$$

$$\hat{\sigma}_v^2 = \hat{\sigma}_x^2 \sin^2 \Theta + \hat{\sigma}_y^2 \cos^2 \Theta - \hat{\sigma}_{xy} \sin(2\Theta) \tag{24}$$

The along- and cross-wind turbulence intensities,  $\hat{T}I_u$  and  $\hat{T}I_v$ , obtained from the variance method, are then defined as:

$$\hat{\mathrm{TI}}_{u} = \frac{\sqrt{\hat{\sigma}_{u}^{2}}}{U} \tag{25}$$

$$\hat{\mathrm{TI}}_v = \frac{\sqrt{\hat{\sigma}_v^2}}{U} \tag{26}$$

# 275 2.7 Velocity spectra

285

290

The velocity power spectral density, hereafter referred to as the velocity spectra, provides valuable insight into the distribution of turbulent kinetic energy across different scales of motion within the wind flow. This information is essential for characterizing turbulence and understanding its impact on wind turbine performance and structural loading.

Velocity spectra were estimated using Welch's method (Welch, 1967), which involves segmenting the time series into overlapping windows, applying a windowing function, computing a periodogram for each segment, and averaging the resulting periodograms to obtain a stable spectral estimate. A Hann window with 50% overlap was used to reduce spectral leakage and improve frequency resolution.

The analysis focused on the along-wind and cross-wind velocity components, derived from time series of the horizontal wind speeds u and v, respectively, as measured by the sonic anemometer. These were used to compute the spectra  $S_{uu}$  and  $S_{vv}$ .

For lidar measurements, two methods were employed to estimate the velocity spectra:

- 1. **Traditional method** Spectra were computed directly from the along-wind and cross-wind velocities, obtained by rotating the instantaneous velocities  $u_x$  and  $u_y$ . This yielded spectral estimates  $\tilde{S}_{uu}$  and  $\tilde{S}_{vv}$ .
- 2. **Variance method** Spectra were computed from the LOS velocity spectra associated with each beam in the 10-beam configuration. These LOS spectra were assembled into the vector  $\beta$ :

$$\beta = [S_{1,A}, S_{2,A}, S_{3,A}, S_{4,A}, S_{5,A}, S_{1,B}, S_{2,B}, S_{3,B}, S_{4,B}, S_{5,B}]$$
(27)

The matrix  $\mathbf{Q}^+$ , defined in Eq. 21, was then used to compute the vector of spectral components:

$$s = \mathbf{Q}^{+} \boldsymbol{\beta} \tag{28}$$

The resulting six-element vector s contains the spectral components associated with the independent elements of the symmetric velocity spectral tensor s, which is reconstructed as:

$$\mathbf{S} = \begin{pmatrix} s_1 & s_4 & s_5 \\ s_4 & s_2 & s_6 \\ s_5 & s_6 & s_3 \end{pmatrix} = \begin{pmatrix} \hat{S}_{xx} & \hat{S}_{xy} & \hat{S}_{xz} \\ \hat{S}_{xy} & \hat{S}_{yy} & \hat{S}_{yz} \\ \hat{S}_{xz} & \hat{S}_{yz} & \hat{S}_{zz} \end{pmatrix}$$
(29)

The along-wind and cross-wind velocity spectra,  $\hat{S}_{uu}$  and  $\hat{S}_{vv}$ , were then obtained via coordinate transformation:

$$\hat{S}_{uu} = \hat{S}_{xx}\cos^2\Theta + \hat{S}_{uy}\sin^2\Theta + \hat{S}_{xy}\sin(2\Theta) \tag{30}$$

$$\hat{S}_{vv} = \hat{S}_{xx}\sin^2\Theta + \hat{S}_{yy}\cos^2\Theta - \hat{S}_{xy}\sin(2\Theta) \tag{31}$$

For simplicity, the along-wind spectra  $S_{uu}$ ,  $\tilde{S}_{uu}$ , and  $\hat{S}_{uu}$ , and cross-wind spectra  $S_{vv}$ ,  $\tilde{S}_{vv}$ , and  $\hat{S}_{vv}$ , are hereafter referred to as  $S_u$ ,  $\tilde{S}_u$ , and  $\hat{S}_u$ , and  $\hat{S}_v$ , respectively, for conciseness.

#### 2.8 Atmospheric stability

305

320

The 30-min subsets were classified into different atmospheric stability regimes based on the Monin–Obukhov length,  $L_{\rm MO}$ , using the thresholds listed in Table 1 (Sathe et al., 2011). The Monin–Obukhov length was estimated via the eddy covariance method (Kaimal and Finnigan, 1994) using 10 Hz measurements from the 3D sonic anemometer. The 30-min virtual potential temperature,  $\theta_{\rm v}$ , was computed by aggregating 10-min averaged temperature measurements from the WXT 530 sensor.  $L_{\rm MO}$  is defined as:

$$L_{\rm MO} = -\frac{u_*^3 \theta_{\rm v}}{\kappa g \sigma_{w \theta_{\rm v}}} \tag{32}$$

where  $\kappa = 0.4$  is the von Kármán constant, g is the acceleration due to gravity,  $\theta_{\rm v}$  is the virtual potential temperature, and  $\sigma_{w\theta_{\rm v}}$  is the covariance between the vertical wind speed w and  $\theta_{\rm v}$ , representing the virtual kinematic heat flux. The friction velocity,  $u_*$ , is computed from the turbulent momentum fluxes as:

$$u_* = \left(\sigma_{uw}^2 + \sigma_{vw}^2\right)^{1/4} \tag{33}$$

where  $\sigma_{uw}$  and  $\sigma_{vw}$  are the covariances between the horizontal velocity components (u and v) and the vertical velocity component w, respectively. Among the 1,098 30-min subsets, 37.1% were recorded under neutral conditions, 20.5% under stable conditions, and 42.4% under unstable conditions. Unless otherwise stated, the results presented in this work are based on the full set of 30-min subsets; results stratified by stability condition are explicitly indicated in the text.

#### 2.9 Error Metrics

To evaluate the performance of TI estimates derived from the lidar using both the traditional and variance methods, two error metrics introduced by DNV are employed: MRBE and RRMSE expressed in percentage (DNV, 2023). The sonic anemometer is used as the reference for all comparisons.

Table 1. Classification of atmospheric stability based on Monin-Obukhov length.

| Stability conditions | $L_{ m MO}$ [m]                  |  |  |  |
|----------------------|----------------------------------|--|--|--|
| Unstable             | $-500 \le L_{\rm MO} \le -50$    |  |  |  |
| Neutral              | $ L_{\mathrm{MO}}  \geq 500$     |  |  |  |
| Stable               | $50 \le L_{\mathrm{MO}} \le 500$ |  |  |  |

The MRBE quantifies the average relative deviation of the lidar-based estimate X from the sonic-based reference  $X_{ref}$ , and is defined as:

MRBE = 
$$\frac{1}{N} \sum_{j=1}^{N} \frac{X_j - X_{\text{ref},j}}{X_{\text{ref},j}}$$
 (34)

The RRMSE reflects the relative magnitude of deviations and is given by:

RRMSE = 
$$\sqrt{\frac{1}{N} \sum_{j=1}^{N} \left(\frac{X_j - X_{\text{ref},j}}{X_{\text{ref},j}}\right)^2}$$
 (35)

Here,  $X_j$  denotes the TI estimated from the lidar (either by the traditional or variance method), and  $X_{ref,j}$  is the corresponding TI derived from the sonic anemometer, for each of the N 30-min ensembles. These metrics provide complementary insight: MRBE indicates systematic bias (over- or underestimation), while RRMSE captures the overall scatter or consistency of the estimate.

# 330 3 Results

#### 3.1 Variances

The cumulative distribution functions (CDFs) of the along- and cross-wind velocity variances, computed from sonic anemometer measurements and lidar-derived estimates using both methods, are presented in Fig. 4. The mean along-wind variance measured by the sonic anemometer,  $\sigma_u^2$ , was found to be 0.721 m<sup>2</sup>/s<sup>2</sup>. For the along-wind variance,  $\tilde{\sigma}_u^2$ , estimated using the traditional method, the mean value was overestimated by 3.5% compared to the reference measurement. In contrast, the variance method yielded an underestimation, with  $\hat{\sigma}_u^2$  being 5.4% lower than the reference mean (Fig. 4a). The cross-wind variance showed a more pronounced discrepancy between the two methods. The CDF of  $\hat{\sigma}_v^2$ , obtained via the variance method, closely matched that of the reference variance,  $\sigma_v^2$ , with nearly identical mean values of 0.474 m<sup>2</sup>/s<sup>2</sup> and a slight underestimation of 0.4% (Fig. 4b). In contrast, the traditional method significantly overestimated the cross-wind variance, with  $\tilde{\sigma}_v^2$  exceeding the reference mean by 33.6%.

To further quantify the differences in variance estimation, we analyzed the slopes of linear regressions between each method's variance estimates and the reference sonic anemometer measurements. As a benchmark, we also include slope results from the 6-beam method reported by Sathe et al. (2015). The results are shown in Fig. 5 and summarized in Table 2. For

conciseness, scatter plots are omitted; however, since the intercepts were consistently near zero, the slope alone effectively characterizes the proportional agreement. Here, slope values (which are dimensionless and typically close to 1) are expressed as percentages by multiplying by 100 to facilitate interpretation—for example, a slope of 0.92 is presented as 92%, indicating that the method estimates 92% of the reference variance.

**Figure 4.** Cumulative distribution functions of the along-wind velocity variances (a) and cross-wind velocity variances (b), computed from sonic anemometer measurements and lidar-derived estimates using the traditional and variance method. The vertical dashed lines shows the mean variances.

**Table 2.** Slope values from linear regressions between wind variances estimated by each method and reference sonic anemometer measurements, categorized by atmospheric stability conditions. "DL" refers to the dual-lidar configuration using the variance method, and "SL" refers to the single-lidar configuration using the traditional method. Results associated with the 6-beam method are reproduced from Sathe et al. (2015). A slope of 1 indicates perfect agreement with the reference.

|                     | Unstable |      |        | Stable |      |        | Neutral |      |        |
|---------------------|----------|------|--------|--------|------|--------|---------|------|--------|
| Methods             | DL       | SL   | 6-beam | DL     | SL   | 6-beam | DL      | SL   | 6-beam |
| Along-wind variance | 0.92     | 1.02 | 0.85   | 0.91   | 0.94 | 0.87   | 0.97    | 1.02 | 1.01   |
| Cross-wind variance | 0.90     | 1.21 | 0.87   | 0.96   | 1.06 | 0.90   | 0.99    | 1.32 | 0.91   |

**Figure 5.** Histograms of the slope values from linear regressions between wind variance estimates and reference sonic anemometer measurements: (a) along-wind variance and (b) cross-wind variance. The comparison includes three methods: the dual-lidar configuration using the variance method, the single-lidar configuration using the traditional method, and the benchmark 6-beam method adapted from Sathe et al. (2015). Dashed areas overlaid on the bars represent the discrepancy between each method's slope and the ideal reference value of 1, which indicates perfect agreement with the reference measurements.

Across all stability regimes, the dual-lidar configuration associated with the variance method consistently estimated between 90% and 97% of the reference variance values for both along- and cross-wind components. Conversely, the single-lidar method associated with the traditional approach generally overestimated variances, particularly the cross-wind component under neutral and unstable conditions, where values reached 132% and 121% of the reference, respectively. Under stable conditions, estimates derived from the traditional approach were closer to the reference, overestimating by 6% or less. The 6-beam method consistently underestimated variance values, with along-wind variances ranging from 85% to 101%, and cross-wind variances from 87% to 91% of the reference, depending on stability.

#### 355 3.2 Velocity spectra

Fig. 6 shows the mean spectra of the along-wind and cross-wind velocity components, averaged under different atmospheric stability conditions and compared with reference spectra obtained from sonic anemometer measurements. Across all regimes and for both components, a consistent pattern emerges: the variance method systematically underestimates spectral energy at low frequencies, agrees reasonably well at intermediate frequencies, and flattens at higher frequencies in a manner characteristic

**Figure 6.** Mean spectra of the along-wind and cross-wind velocity components derived from the traditional and variance method, averaged under different atmospheric stability conditions, and compared against reference spectra derived from sonic anemometer measurements.

of white noise. Overall, however, the mean spectra from the variance method remain largely consistent with the reference spectra.

In contrast, the traditional method reproduces the reference spectra more accurately at low frequencies. For the along-wind component, it matches the reference spectra under neutral and stable conditions (Fig. 6b–c). Under unstable conditions, however, the along-wind spectrum (Fig. 6a), as well as the cross-wind spectra across all stability regimes (Fig. 6d–f), consistently exceed the reference spectra at intermediate and higher frequencies. This leads to a systematic overestimation of spectral energy and variance.

### 3.3 Turbulence intensity

365

Fig. 7a-c present scatter plots of the along- and cross-wind TI estimates,  $\tilde{\mathrm{TI}}_u$  and  $\tilde{\mathrm{TI}}_v$ , obtained using the traditional method, plotted against the reference TI values. In contrast, Fig. 7b-d display scatter plots of the along- and cross-wind TI estimates,  $\hat{\mathrm{TI}}_u$  and  $\hat{\mathrm{TI}}_v$ , derived using the variance method also plotted against the reference TI. Linear regression models fitted to each dataset are constrained to pass through the origin (zero intercept). The slopes of these regression lines reveal that the traditional method systematically overestimates TI, with  $\tilde{\mathrm{TI}}_u$  and  $\tilde{\mathrm{TI}}_v$  being approximately 107% and 118% of the reference values, respectively. Conversely, the variance method underestimates TI, yielding  $\hat{\mathrm{TI}}_u$  and  $\hat{\mathrm{TI}}_v$  values that correspond to approximately 95% and 97% of the reference TI. In terms of coefficient of determination (R<sup>2</sup>), the variance method demonstrates improved

Figure 7. Scatter plots of along- and cross-wind TI estimates obtained using two different methods. Panels (a) and (c) show  $\tilde{T}I_u$  and  $\tilde{T}I_v$ , respectively, derived using the traditional method applied to single-lidar measurements. Panels (b) and (d) show  $\hat{T}I_u$  and  $\hat{T}I_v$ , respectively, derived using the variance method based on dual-lidar measurements. Red dashed lines indicate linear regression fits, with the corresponding regression equations and  $R^2$  values displayed in the bottom-right corner of each panel. Points are color-coded according to normalized kernel density estimation (KDE).

performance for the along-wind component, achieving  $R^2 = 0.94$ , compared to  $R^2 = 0.90$  for the traditional method. However, for the cross-wind component, the traditional method slightly outperforms the variance method, with  $R^2 = 0.94$  versus  $R^2 = 0.90$ .

Fig. 8 shows the binned-averaged MRBE and RRMSE of the along-wind and cross-wind TI as a function of binned-averaged wind speed. A key observation is that the error metrics associated with the reconstruction of TI using the variance method are consistently lower than those from the traditional method across nearly all wind speed bins. There is no clear evidence of a strong dependence of MRBE or RRMSE on wind speed, except for the RRMSE of the along-wind TI (Fig. 8b), which clearly

decreases as wind speed increases. The absolute MRBE for along-wind TI derived using the variance method remains within the range of 0-5%, while this range nearly doubles (0-10%) for the traditional method. For both methods, the error associated with cross-wind TI is approximately 1.5 to 1.8 times higher than the respective error for along-wind TI. Furthermore, the error metrics for the traditional method generally vary over a range nearly twice as wide as those for the variance method.

Fig. 9 shows the mean absolute MRBE and RRMSE for along- and cross-wind TI reconstructed using both methods. The results are summarized in Table 3. The mean absolute MRBE values for along-wind TI indicate that the variance method yields lower errors across all stability regimes. Specifically, in unstable conditions, the MRBE decreases from 10.4% using

**Figure 8.** Binned-averaged MRBE (a-c) and RRMSE (b-d) of the along-wind and cross-wind TI derived from the traditional and variance methods as a function of binned-averaged wind speed. Only bins with sufficient data are shown.

**Table 3.** Comparison of mean absolute MRBE and RRMSE for along-wind and cross-wind TI estimates reconstructed using the traditional method (TM) with single-lidar (SL) measurements and the variance method (VM) with dual-lidar (DL) measurements, across different atmospheric stability conditions. The category "All" represents the full dataset without separation by stability class.

| Conditions                | Unstable |          | St      | able     | Neutral |         | All     |         |
|---------------------------|----------|----------|---------|----------|---------|---------|---------|---------|
| Methods                   | SL — TM. | DL — VM. | SL — TM | DL — VM. | SL — TM | DL - VM | SL — TM | DL — VM |
| MRBE  (%) — Along-wind TI | 10.4     | 7.4      | 8.3     | 8.1      | 7.3     | 6.3     | 10.1    | 7.5     |
| MRBE  (%) — Cross-wind TI | 20.0     | 11.4     | 8.0     | 9.2      | 18.8    | 13.4    | 17.6    | 12.3    |
| RRMSE (%) — Along-wind TI | 18.1     | 11.7     | 11.0    | 10.6     | 9.8     | 9.2     | 19.5    | 11.7    |
| RRMSE (%) — Cross-wind TI | 24.8     | 15.5     | 13.1    | 14.6     | 24.1    | 19.0    | 23.4    | 18.2    |

Figure 9. Mean absolute MRBE (a) and RRMSE (b) for along-wind (solid colored bars) and cross-wind (patterned colored bars) TI estimates reconstructed using the traditional method and the variance method across different atmospheric stability conditions. The "All" category represents the entire dataset without differentiation by stability class.

the traditional method to 7.4% using the variance method. In stable and neutral conditions, the variance method maintains slightly lower MRBE values of 8.1% and 6.3%, compared to 8.3% and 7.3% for the traditional method, respectively. For the entire dataset, the mean absolute MRBE for the variance method is 7.5%, compared to 10.1% for the traditional method. For

cross-wind TI, the traditional method yields higher MRBE values under unstable and neutral conditions (20.0% and 18.8%, respectively), while the variance method reduces these errors to 11.4% and 13.4%. Under stable conditions, the traditional method produces a slightly lower MRBE than the variance method (8.0% versus 9.2%). Considering the full dataset, the mean absolute MRBE decreases from 17.6% with the traditional method to 12.3% with the variance method.

Regarding RRMSE, for along-wind TI, the variance method results in lower errors than the traditional method for all stability regimes. Across the entire dataset, the variance method achieves an RRMSE of 11.7%, whereas the traditional method reports 19.5%. For cross-wind TI, the variance method lowers RRMSE values during unstable and neutral conditions but produces a slightly higher error under stable conditions. Across the full dataset, however, the variance method achieves a lower overall RRMSE of 18.2% compared to 23.4% for the traditional method.

### 4 Discussion

The ability of the variance method applied in a dual-lidar configuration to measure turbulence was compared against both the traditional approach employed in the wind power industry and the 6-beam method, with results from the latter taken from Sathe et al. (2015). Since the 6-beam method was evaluated using a completely different dataset, it is not possible to identify a single best method for turbulence estimation. Instead, the goal here is to provide an order-of-magnitude comparison and highlight possible improvements.

As with the variance method, the 6-beam approach is affected only by intra-beam effect. Although the authors did not explicitly remove instrumental noise, its contribution was considered negligible. The 6-beam method relies on measurements from the WindScanner, which incorporates the scanning pulsed lidar WindCube 200 by Vaisala (Vasiljević et al., 2016). This device features an extended probe length of about 100 m, in contrast to pulsed lidar profilers. Since noise levels decrease with increasing probe volume, the WindScanner measurements are therefore expected to be largely unaffected by noise. In general, smaller probe volumes produce higher noise because fewer scatterers contribute to the backscattered signal, which reduces the signal strength relative to the detector's noise floor. Consequently, the signal-to-noise ratio decreases, resulting in higher instrumental noise in the retrieved measurements.

Overall, the 6-beam method produced lower variances compared to the variance method applied on the dual-lidar configuration, which itself tends to underestimate turbulence metrics. This discrepancy is most likely attributable to the WindScanner's lower sampling rate, which is nearly four times smaller than that of the WindCube v2.1. The sampling rate is governed by the accumulation time,  $\Delta t$ , which enters the transfer function, H, applied by the instrument to the measured signal (Thiébaut et al., 2025):

$$|H|^2(\mathbf{k}) = \operatorname{sinc}^2\left(\frac{\Delta t}{2}\mathbf{k} \cdot \mathbf{U}\right) \exp\left(-\left[\sigma_l^2(\mathbf{k} \cdot \mathbf{b})^2 + \sigma_r^2(\|\mathbf{k}\|^2 - (\mathbf{k} \cdot \mathbf{b})^2)\right]\right)$$
 (36)

Here,  $\mathbf{k}$  is the turbulent structure wavevector,  $\mathbf{b}$  is the beam pointing vector,  $\mathbf{U}$  is the wind velocity vector of magnitude U, and  $\sigma_l$  and  $\sigma_r$  represent Gaussian weighting factors in the along-beam and cross-beam directions, respectively. From Eq. 36,

it follows that wind field structures with wavelengths smaller than  $\sigma_l$  in the along-beam direction are attenuated, as are those with wavelengths smaller than  $\sigma_r$  in the cross-beam direction. However, in the latter case, these structures are so small that the filtering effect becomes negligible, since the probe cross-section is approximately 1 cm (Thiébaut et al., 2025). Assuming the Taylor frozen turbulence hypothesis, the wavevector domain transmitted by the filter is defined by the intersection of two slices: one perpendicular to U, which preserves structures longer than  $\pi \Delta t U$ , and another perpendicular to b, which retains structures longer than  $\sigma_l$ . All other structures are filtered out.

To preserve smaller structures,  $\Delta t$  must be reduced, thereby increasing the sampling rate. Thiébaut et al. (2025) demonstrated that raising the sampling rate from 0.25 Hz (commercial lidar) to 1 Hz for the WindCube v2.1 increases the along-wind variance derived from the variance method by about 7% compared with the commercial sampling rate, without significantly reducing data availability (a loss of 0.5%). A target sampling rate of 1 Hz for future lidar profilers may therefore represent a good trade-off between data availability and turbulence resolution.

The 6-beam method forms the basis of the pulsed Beam6X WindPower lidar profiler developed by Lumibird, which combines five slanted beams with one vertical beam. The variance method described in this paper can be applied to this technology using a single lidar. However, a full measurement cycle over the six beams of the Beam6X WindPower requires approximately 6 s, corresponding to a sampling rate of 0.167 Hz—only 33% of that of the commercial WindCube v2.1 and 83% lower than the targeted 1 Hz. Consequently, the turbulence measurement performance of the Beam6X WindPower is currently expected to be comparable to that of the WindScanner. Reducing the accumulation time and thus increasing the sampling rate will likely be necessary to match the performance of the dual-lidar configuration with two WindCube v2.1 systems. Nevertheless, the clear advantage of the Beam6X WindPower is that the variance method becomes applicable in all wind directions with a single lidar.

The variance method can be applied with the continuous-wave ZX300 lidar profiler. The ZX300 records about 50 LOS positions per second, corresponding to an accumulation time of 20 ms per LOS, which increases measurement noise. These LOS positions are distributed azimuthally along a circle at a fixed elevation angle. A further drawback is that a full scan at one altitude requires 1 s, meaning that the effective sampling rate decreases with the number of altitudes measured. To reach a 1 Hz sampling rate, the device must therefore be restricted to a single altitude, most likely the hub height of wind turbines. To reproduce the 0.25 Hz sampling rate of the commercial WindCube v2.1, measurements could be taken at a maximum of four altitudes. Despite these constraints, the ZX300 offers the advantage of a shorter probe length that increases with altitude (e.g., 7.7 m at 100 m above ground level), compared to the WindCube v2.1, which has a constant probe length of 23 m (Thiébaut et al., 2025). The shorter probe length has the potential to reduce intra-beam averaging errors and thereby improve the accuracy of turbulence estimates.

The variance method produces lower mean MRBE and RRMSE for all stability conditions except for the cross-wind TI during stable conditions. During these conditions, turbulence is generally weak and dominated by larger, more coherent eddies. In this regime, the assumption of spatial homogeneity is more likely to hold, so inter-beam contamination is limited and the traditional method tends to yield smaller errors in TI reconstruction. In contrast, under unstable or neutral conditions, turbulence is stronger, more intermittent, and characterized by a broader range of eddy sizes. Violations of the homogeneity assumption

become more pronounced, enhancing inter-beam contamination in the traditional method. As the variance method is not subject to inter-beam effects and relies directly on LOS velocity statistics, it provides more reliable estimates of TI in unstable and neutral regimes, despite remaining sensitive to intra-beam averaging.

The inter-beam effect, inherent only to the traditional method, is the most unpredictable because it depends strongly on local site characteristics. Physically, this effect arises when different beams of the scanning pattern probe air volumes that are not dynamically coherent. For example, in complex terrain the airflow may accelerate over a ridge or decelerate in a sheltered valley. Beams sampling these distinct regions can therefore register artificially large velocity differences that do not reflect true turbulence but rather systematic spatial gradients in the mean flow. Similarly, surface roughness or obstacles (e.g., trees, buildings) can generate localized shear layers and recirculation zones, adding spurious high-frequency variability to some beams while leaving others unaffected. Such effects can either amplify or attenuate the apparent turbulent energy when the beams are combined, making error metrics highly site-specific and difficult to generalize.

In contrast, the variance method is impacted primarily by the intra-beam effect, which is systematic and thus more predictable. This makes the variance method potentially more robust and transferable across different sites. Furthermore, TI derived from the variance method exhibited smaller variations across the stability regimes examined, suggesting that atmospheric stratification has a weaker influence on its ability to capture the underlying turbulence level compared to the traditional method.

For both methods, a marked difference was observed between the along-wind and cross-wind TI, with significantly larger errors associated with the latter. Nevertheless, the variance method consistently yielded lower errors than the traditional approach. This difference is mainly due to the rotation from instrument coordinates to along- and cross-wind coordinates: the along-wind component is relatively robust to small directional uncertainties, while the cross-wind component is much more sensitive. Even small errors in the estimated wind direction can lead to disproportionately large errors in the cross-wind variance. In addition, because turbulence energy is generally higher in the along-wind than in the cross-wind direction, the relative impact of these directional errors becomes more pronounced for the latter. Together, these factors explain why the along-wind variance is typically well captured, whereas the cross-wind variance remains more uncertain.

This imbalance between the along- and cross-wind variances is not only a methodological concern but also has practical implications for wind turbine load assessment. The along-wind component dominates TI and is directly linked to variations in power production, as well as to fatigue and ultimate loads on the rotor blades and tower. In contrast, the cross-wind component plays a critical role in generating yaw and lateral tower loads, and in inducing asymmetric blade loading. Despite the higher uncertainty in cross-wind variance, the variance method still outperforms the traditional method in capturing this component, reducing measurement errors and improving overall turbulence characterization. Although its variance is typically smaller, underestimating the cross-wind turbulence leads to a systematic bias in the predicted lateral excitation of the turbine. As a result, the use of the variance method without accounting for this limitation may yield non-conservative estimates of turbine loads, especially for large rotors where yaw misalignment and lateral inflow fluctuations are increasingly significant.

#### 490 5 Conclusions

The variance method applied in a dual-lidar configuration demonstrates improved performance compared to the traditional method in terms of MRBE and RRMSE for both along- and cross-wind TI. Because it is primarily affected by the intra-beam effect, which is systematic and predictable, the variance method is largely site-independent and broadly applicable, thereby overcoming the main limitation of the traditional method associated with unpredictable inter-beam errors. Moreover, turbulence estimates derived from the variance method remain consistent across varying atmospheric stability conditions, highlighting its resilience to stratification effects. Taken together, these features indicate that the dual-lidar approach is now sufficiently mature for validation and practical deployment in ground-based wind measurements.

Cross-wind turbulence exhibits higher relative uncertainty than along-wind turbulence because of its sensitivity to wind direction estimation. Nonetheless, the variance method captures the overall turbulence characteristics accurately. This distinction has practical relevance: along-wind turbulence, which dominates power production and rotor fatigue, is well resolved, while the cross-wind component, though smaller, remains adequately represented to inform assessments of lateral loading.

Further improvements in lidar technology and scanning strategies can enhance the performance of the variance method. Its main limitations with current profilers are the short accumulation time per LOS and the reduced effective sampling rate when scanning multiple altitudes. Nevertheless, the variance method remains a promising and versatile approach for accurate, site-independent turbulence measurements, supporting reliable wind energy assessments and load modeling.

#### **Author contributions**

MT identified the problematic, performed the analysis and drafted the paper. NL reviewed the manuscript.

# Data availability

The high-frequency velocity data from the dual-lidar and sonic anemometer measurements used in this study are provided alongside this paper. To comply with confidentiality agreements, all velocity components have been multiplied by an undisclosed constant scaling factor. This transformation preserves all relative variations, turbulence characteristics, and statistical relationships among instruments.

### Acknowledgments

We would like to thank Vaisala for providing the secondary lidar, which enabled the implementation of the dual-lidar configuration and made this study possible.

# **Competiting interest**

The authors declare that they have no conflict of interest.

# **Financial support**

This work was made possible through the support of France Energies Marines and the French government, managed by the
520 Agence Nationale de la Recherche under the Investissements d'Avenir program, with the reference ANR-10-IEED-0006-34.

This work was carried out in the framework of the DRACCAR-NEMO project.

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
