# Peer review of "Dual-lidar profilers for measuring atmospheric turbulence"

_Wind Energy Science, 2025_

## Referee Comment (RC1)

**Review of Thiébaut and Luxcey "Dual-lidar profilers for measuring atmospheric turbulence" submitted to Wind Energy Science**

Jakob Mann

December 2025

**1 General comment**

This paper investigates the line-of-sight variance method to extract turbulence from profiling lidars. The authors cleverly use two standard five-beam pulsed Doppler lidars rotated 45° relative to each other to essentially for a nine-beam lidars. This allows of obtaining the variances of the horizontal velocities without combining line-of-sight velocities instantaneously from different beams, similarly to how it is done in Sathe, Mann, et al. (2015).

The method is tested with relevant offshore data where the turbulence estimations from the lidars are compared to sonic anemometer "ground truth". The results are promising, but the impact of measurement volume averaging is still not successfully addressed.

In general, the paper is publishable, but there are a number of comments that has to be addressed, and the paper is at places too long and textbook-like.

**2 Specific comments**

l 65 – 69 The considerations here are not entirely correct. If you measure with an instrument with point-like measurement volume and high time-resolution, you will get the real turbulence variance. If you pick only one sample every second, you still get the correct turbulence variance, so you resolve all scales. Similarly, for the six-beam WindScanner setup it is not he 15 $s$ that is important. It is the sample volume and the averaging time of the individual beam that determined what scales contribute to the variance. Had the sampling volume and the averaging time been small, then the variance would have been unbiased, even if the cycle were completed in 15 $s$.

l 93 *Aldernay Race* sounds like a ship race, but it is a geographic location. Maybe you could help the reader realize that.

l 140 This comment relates to the first. There is no reason to downsample the cup anemometer for a consistent comparison. I think what is most relevant to compare with the the variance of the full cup anemometer signal. It is also unclear what the low-pass filter is doing in this comparison. Please specify the low-pass exactly, and why it is applied.

l 149 The choice of 30-min versus 10-min averages is very important. I think it is very reasonable, but it is not in the DNV error metrics. This point deserves some more emphasis.

l 162 Aerosol fall speed? Do you mean rain?

l 176 A similar technique was appied by Mayor et al. (1997). Cite, if you see fit.

sec 2.6.1 This section is too long and text book like. Please short considerable. Why is $\text{TI}_u = \sqrt{\sigma_u^2}/U$ and not just $\text{TI}_u = \sigma_u/U$? Same for Eqs 25+26.

Eqs 16 + 17 Is this really how the variances are calculated? First you calculate the spectrum and then integrate over frequencies.

Eq 18 + 19 There is something wrong with the notation (I don't think anything is basically wrong with the math). In 18, a 10 by 3 matrix is multiplied by a 3 by 3 matrix. This gives a 10 by 3 matrix. That is added to a 10 by 3 matrix, giving again a 10 by 3 matrix. It is unclear how that turnrs into a 10 by 6 matrix in 19. In 19 the LHS is a matrix, but you refer to the elements (e.g $Q_{q,m}$?). Are you summing over repeated indices? Please clean up the notation for easier reading.

l 319 The abbreviations MRSE and RRMSE were defined in the abstract. Maybe it would be helpful to do it again here.

l 345 Why not simply force the fits through zero? (late, in l 371 you actually state that, which I think is good)

l 346 I disagree that a percentage facilitates interpretation. Could be omitted.

Figure 4 I'm not sure this figure is necessary. It could be omitted to reduce the length of an already too long paper.

Tab 2, Fig 5 I think it is very difficult to compare experiments that have been performed in different climates, different instruments, and different beam geometry.

Fig 6 Excellent, but I insist that you plot premultiplied spectra $f \cdot S(f)$ when you have a log frequency axis. Explain how the spectra are average. Just plain averaging or averages of $S/U^2$ or $S/\sigma^2$? The latter has the advantage that one very strong wind case does not dominate everything.

l 358   It is surprising that the velocity spectra derived with the variance method do not match at low frequencies, while the traditional indeed do match. Can this be explained? In Sathe and Mann (2012) you see good match at low frequencies for the traditional method for the $u$- and $w$-components, but not for the $v$-component. It is also a bit strange that the overestimation of the traditional along-wind spectra differ at high frequencies with stability. Can that be explained?

Figure 7   Not because I want it in the paper, but you do similar analysis of $TI_w$?

Figure 8   Please do not show $|MRBE|$. A bias should be only shown with its sign.

Discussion   A very good discussion i, in general!

l 410   In Sathe, Mann, et al. (2015) it is stated that the WindScanner system can have either 400 or 200 ns pulses. Although not entirely clear from the paper, the 200 ns pulse was used. That corresponds to a FWHM probe volume of approximately 30 m, not 100 m, as stated in the text.

l 417   Relating to previous comments, the sample rate in it self should not bias the variance. Only the probe volume and the accumulation time should have an impact.

Eq 36   It is unnecessary to include the $\sigma_r^2$ term in the equation. It is completely negligible for a lidar, and only confuses the reader.

l 422   $\sigma_l$ and $\sigma_r$ are not weighting factors, but length scales.

l 436   Again, it should be the averaging, not the sample rate that is important.

$\approx$ l 411   It think it is worth mentioning Manami et al. (2025) in the discussion. In this paper we try to annihilate the probe volume effect of at pulsed lidar.

l 443 – 452   Again, if the ZXLidar is taking 50 ms acculation time, then that, together with the probe volume, is what is important. Spending only a fraction of the time at one height does not introduce a bias in the variance. That is actually also discussed in Lenschow, Mann, and Kristensen (1994).

l 453 – 457   I don't understand this discussion. Under stable conditions, the standard knowledge is that the length scale is *smaller* for stable conditions.

l 465   How can systematic (that is stationary) spatial gradients introduce more variance in the traditional method? I would think that it only introduces a bias in the mean.

473 – 476   Interesting that the error is significantly larger for the cross-wind component, but can you explain why? I cannot really follow the logic in the explanation. Maybe there is a hint in Sathe and Mann (2012).

l 507   "identified the problematic". I guess you mean "problem", or "research issue".

**References**

Manami, M., J. Mann, M. Sjöholm, G. Léa, and G. Gorju (2025). "Squeezing turbulence statistics out of a pulsed Doppler lidar". In: *Atmospheric Measurement Techniques* 18.23, pp. 7513–7523. DOI: 10.5194/amt-18-7513-2025. URL: https://amt.copernicus.org/articles/18/7513/2025/.

Sathe, A., J. Mann, N. Vasiljevic, and G. Lea (2015). "A six-beam method to measure turbulence statistics using ground-based wind lidars". In: *Atmospheric Measurement Techniques* 8.2, pp. 729–740.

Sathe, A. and J. Mann (2012). "Measurement of turbulence spectra using scanning pulsed wind lidars". In: *Journal of Geophysical Research: Atmospheres* 117.D1.

Mayor, S. D., D. H. Lenschow, R. L. Schwiesow, J. Mann, C. L. Frush, and M. K. Simon (1997). "Validation of NCAR 10.6-$\mu$ m CO 2 Doppler lidar radial velocity measurements and comparison with a 915-MHz profiler". In: *Journal of Atmospheric and Oceanic Technology* 14.5, pp. 1110–1126.

Lenschow, D., J. Mann, and L. Kristensen (1994). "How long is long enough when measuring fluxes and other turbulence statistics?" In: *Journal of Atmospheric and Oceanic Technology* 11.3, pp. 661–673.

---

## Author Comment (AC1)

**Response to Reviewer 1**

Manuscript Title: *Dual-lidar profilers for measuring atmospheric turbulence*
Manuscript ID: wes-2025-179
Journal: *Wind Energy Science*

Dear Jacob Mann,

We sincerely thank you for your careful reading of our manuscript and your comments. We have revised the manuscript accordingly. Below, we provide a point-by-point response to all comments. Your comments are shown in black, and our responses are shown in blue. Changes in the manuscript are highlighted or tracked as requested by the journal.

**Reviewer 1**

**General comment**

This paper investigates the line-of-sight variance method to extract turbulence from profiling lidars. The authors cleverly use two standard five-beam pulsed Doppler lidars rotated 45∘ relative to each other to essentially for a nine-beam lidars. This allows of obtaining the variances of the horizontal velocities without combining line-of-sight velocities instantaneously from different beams, similarly to how it is done in Sathe et al. (2015).

The method is tested with relevant offshore data where the turbulence estimations from the lidars are compared to sonic anemometer "ground truth".

The results are promising, but the impact of measurement volume averaging is still not successfully addressed. In general, the paper is publishable, but there are a number of comments that has to be addressed, and the paper is at places too long and textbook-like.

**Response:** Thank you for the positive overall assessment of the manuscript and for recognizing the novelty of the experimental configuration and the potential of the variance method for turbulence estimation from lidar profilers. We acknowledge your concern regarding the impact of measurement volume (probe-time and probe-length) averaging, which remains a fundamental limitation for turbulence retrievals from pulsed Doppler lidars. This limitation is now more clearly stated and discussed in the revised manuscript, including reference to recent complementary approaches aiming to mitigate probe-volume effects (e.g. Manami et al. (2025)). We also acknowledge your remark that parts of the manuscript were overly textbook-like, particularly in the methodology section. This section has been carefully revised and streamlined to reduce general background material, improve conciseness, and focus more directly on the specific implementation and assumptions relevant to the present study. As a result of these revisions, the total length of the manuscript has been reduced by approximately five pages. All other comments have been addressed in detail below.

**Specific comments**

**Comment 1:** l 65 – 69. The considerations here are not entirely correct. If you measure with an instrument with point-like measurement volume and high time-resolution, you will get the real turbulence variance. If you pick only one sample every second, you still get the correct turbulence variance, so you resolve all scales. Similarly, for the six-beam WindScanner setup it

is not he 15 s that is important. It is the sample volume and the averaging time of the individual beam that determined what scales contribute to the variance. Had the sampling volume and the averaging time been small, then the variance would have been unbiased, even if the cycle were completed in 15 s.

**Response:** This passage has been removed in the revised manuscript, as it was not central to the main objectives (and also wrong) of the study and contributed unnecessarily to the length of the paper.

**Comment 2:** l 93. *Aldernay Race* sounds like a ship race, but it is a geographic location. Maybe you could help the reader realize that.

**Response:** The text has been revised to clarify that the Alderney Race is a geographic location (a tidal channel), in order to avoid confusion for the reader. p.3, l.79.

**Comment 3:** l 140. This comment relates to the first. There is no reason to downsample the cup anemometer for a consistent comparison. I think what is most relevant to compare with the the the variance of the full cup anemometer signal. It is also unclear what the low-pass filter is doing in this comparison. Please specify the low-pass exactly, and why it is applied.

**Response:** The text has been revised to remove the physical interpretation associated with the downsampling. The downsampling of the sonic anemometer data is now described only as a practical step for point-by-point temporal comparison with the lidar measurements, without implying that it is required for a consistent variance comparison. The discussion of turbulence scales and low-pass filtering has been removed accordingly. p.5, l.122-124.

**Comment 4:** l 149. The choice of 30-min versus 10-min averages is very important. I think it is very reasonable, but it is not in the DNV error metrics. This point deserves some more emphasis.

**Response:** This point has been emphasized in the revised manuscript. The text now explicitly notes that the 30-min averaging window differs from the 10-min intervals used in DNV error metrics and clarifies that this choice was made deliberately to improve the statistical convergence of turbulence measurements. p.6, l.129-132.

**Comment 5:** l 162. Aerosol fall speed? Do you mean rain?

**Response:** The text has been revised to clarify that this refers to the motion of aerosol tracers and not to precipitation or rain. p.7, l.142-143.

**Comment 6:** l 176. A similar technique was applied by Mayor et al. (1997). Cite, if you see fit.

**Response:** Thank you for pointing out this study. The reference to Mayor et al. (1997) has been added to the manuscript at the relevant location. p.8, l.156.

**Comment 7:** sec 2.6.1. This section is too long and text book like. Please short considerable. Why is $TI_u = \sqrt{\sigma_u^2}/U$ and not just $TI_u = \sigma_u/U$? Same for Eqs 25+26.

**Response:** Section 2.6.1 has been considerably shortened by removing textbook-style derivations and repeated definitions, while retaining only the equations and descriptions required for the present analysis. In addition, the turbulence intensity notation has been clarified by defining $\sigma_u$ and $\sigma_v$ as standard deviations rather than variances. As a result, the turbulence intensities are now consistently expressed as $TI_u = \sigma_u/U$ and $TI_v = \sigma_v/U$, and Eqs. 13-14 and

25–26 have been removed. p.8-9, l.170-179, Eq. 3-4.

**Comment 8:** Eqs 16 + 17. Is this really how the variances are calculated? First you calculate the spectrum and then integrate over frequencies.

**Response:** The text has been revised to clarify this point. In the present analysis, LOS velocity variances are computed directly from the time series after noise correction. Variances obtained from spectral integration were also evaluated and were found to be equivalent. However, as no explicit spectral integration is required for the results presented in the manuscript, the spectral formulation could be misleading and has therefore been removed for clarity.

**Comment 9:** Eqs 18 + 19. There is something wrong with the notation (I don't think anything is basically wrong with the math). In 18, a 10 by 3 matrix is multiplied by a 3 by 3 matrix. This gives a 10 by 3 matrix. That is added to a 10 by 3 matrix, giving again a 10 by 3 matrix. It is unclear how that turns into a 10 by 6 matrix in 19. In 19 the LHS is a matrix, but you refer to the elements (e.g $Q_{q,m}$). Are you summing over repeated indices? Please clean up the notation for easier reading.

**Response:** The notation has been revised for clarity. The transformation matrix $\mathbf{T}$ is now explicitly defined as a $10 \times 3$ matrix, and the construction of the $10 \times 6$ matrix $\mathbf{Q}$ is clarified by defining it row-wise from quadratic combinations of the elements of $\mathbf{T}$. This makes explicit how the LOS variance vector is related to the six independent components of the Reynolds stress tensor and removes ambiguity regarding matrix dimensions, index usage, and implicit summation, while preserving the original mathematical formulation. p.9-10, Eq. 6-9.

**Comment 10:** l 319. The abbreviations MRSE and RRMSE were defined in the abstract. Maybe it would be helpful to do it again here.

**Response:** The abbreviations MRSE and RRMSE are now redefined at their first occurrence in the main text (introduction section) to improve clarity. p.3, l.87-88.

**Comment 11:** l 345. Why not simply force the fits through zero? (late, in l 371 you actually state that, which I think is good)

**Response:** All material related to this comment has been removed, as Section 3.1 (Variances) has been deleted in the revised manuscript for conciseness and relevance.

**Comment 12:** l 346. I disagree that a percentage facilitates interpretation. Could be omitted.

**Response:** All material related to this comment has been removed following the deletion of Section 3.1 (Variances) in the revised manuscript.

**Comment 13:** Fig. 4. I'm not sure this figure is necessary. It could be omitted to reduce the length of an already too long paper.

**Response:** Fig. 4 has been removed together with Section 3.1 (Variances), which has been deleted in the revised manuscript to reduce length and improve focus.

**Comment 14:** Tab. 2, Fig. 5. It is very difficult to compare experiments that have been performed in different climates, different instruments, and different beam geometry.

**Response:** The discussion and results associated with this comment have been removed following the deletion of Section 3.1 (Variances). The revised manuscript now avoids such

cross-experimental comparisons.

**Comment 15:** Fig. 6. Excellent, but I insist that you plot premultiplied spectra $f \cdot S(f)$ when you have a logarithmic frequency axis. Explain how the spectra are averaged.

**Response:** Fig. 6 (now Fig. 5) has been revised to show premultiplied spectra $f \cdot S(f)$. In addition, the description of the spectral averaging procedure has been clarified and is now explicitly stated in the figure title. p.14, Fig. 5.

**Comment 16:** l 358. It is surprising that the velocity spectra derived with the variance method do not match at low frequencies, while the traditional indeed do match. Can this be explained? In Sathe and Mann (2012) you see good match at low frequencies for the traditional method for the $u$- and $w$-components, but not for the $v$-component. It is also a bit strange that the overestimation of the traditional along-wind spectra differ at high frequencies with stability. Can that be explained?

**Response:** Thank you for the comment. The different low-frequency behaviour arises from the fundamentally different nature of the two retrieval methods. The variance method is based on LOS velocity variance within the lidar probe volume and is therefore less sensitive to large-scale, spatially coherent motions, which mainly affect the mean LOS velocity and contribute only weakly to its variance. This leads to a systematic underestimation of low-frequency spectral energy when compared with point measurements from the sonic anemometer. In contrast, the traditional method reconstructs instantaneous velocity components and therefore retains sensitivity to large-scale motions, resulting in good low-frequency agreement with the sonic, consistent with the findings of Sathe and Mann (2012). The reduced performance for the $v$-component reported in that study is attributed to limitations imposed by the scanning geometry. The high-frequency overestimation observed for the traditional method is caused by the wavenumber-dependent response associated with beam separation (Kelberlau et al., 2020). This effect is strongest under neutral and unstable conditions, when the inertial subrange is well developed, and is weaker under stable stratification where small-scale turbulence is suppressed. These explanations have now been clarified in the revised manuscript. p.18-19, l.365-381.

**Comment 17:** Fig. 7. Not because I want it in the paper, but you do similar analysis of $\mathrm{TI}_w$?

**Response:** No, a similar analysis of $\mathrm{TI}_w$ was not performed, either within this manuscript or as separate side work.

**Comment 18:** Fig. 8. Please do not show |MRBE|. A bias should be shown with its sign.

**Response:** Fig. 8 has been revised to remove |MRBE|. The bias is now shown with its sign.

**Comment 19:** Discussion. A very good discussion in general!

**Response:** Thank you for this positive comment !

**Comment 20:** l 410. In Sathe et al. (2015) it is stated that the WindScanner system can have either 400 or 200 ns pulses. Although not entirely clear from the paper, the 200 ns pulse was used. That corresponds to a FWHM probe volume of approximately 30 m, not 100 m, as stated in the text.

**Response:** The description of the WindScanner pulse duration and the associated probe

length has been removed from the revised manuscript, as these details are no longer required for the present analysis.

**Comment 21:** l 417. Relating to previous comments, the sample rate in it self should not bias the variance. Only the probe volume and the accumulation time should have an impact.

**Response:** Thank you for this clarification. The passage referring to a potential influence of the sampling rate has been removed from the revised manuscript. The discussion now makes clear that the intra-beam effect is governed by probe-volume averaging, specifically the accumulation time and the probe length, rather than by the sampling frequency itself.

**Comment 22:** Eq. 36. It is unnecessary to include the $\sigma_r^2$ term in the equation. It is completely negligible for a lidar, and only confuses the reader.

**Response:** Thank you for this comment. Eq. 36 has been removed from the revised manuscript, as it is no longer required for the presentation of the method. Consequently, the $\sigma_r^2$ term is no longer included.

**Comment 23:** l 422. $\sigma_r$ and $\sigma_l$ are not weighting factors, but length scales.
**Response:** The equation has been removed. See previous comment.

**Comment 24:** l 436. Again, it should be the averaging, not the sample rate that is important.

**Response:** We have revised the text to emphasize that the relevant controlling parameter is the accumulation (averaging) time at each LOS position, rather than the nominal sampling rate. References to sampling rate have been reformulated or removed where appropriate, and the discussion now consistently focuses on the role of temporal averaging in filtering turbulent fluctuations. p.17, l.339-346.

**Comment 25:** $\sim$ 411. It think it is worth mentioning Manami et al. (2025) in the discussion. In this paper we try to annihilate the probe volume effect of a pulsed lidar.

**Response:** Thank you for this very relevant suggestion. We have now explicitly included a discussion of Manami et al. (2025) in the manuscript. Your work is cited in the context of probe-time and probe-volume averaging effects in pulsed Doppler lidars. We clarify that, while the present study investigates how turbulence retrieval is affected by accumulation time, probe length, and scanning configuration, Manami et al. (2025) propose a complementary signal-level approach that aims to mitigate (or "annihilate") probe-volume filtering by exploiting Doppler spectral information. This addition places our results in the broader context of recent efforts to recover turbulence statistics from pulsed lidars and highlights the complementarity between configuration-based and signal-processing-based strategies. p.17, l.347-354.

**Comment 26:** l 443-452. Again, if the ZXLidar is taking 50 ms acculation time, then that, together with the probe volume, is what is important. Spending only a fraction of the time at one height does not introduce a bias in the variance. That is actually also discussed in Lenschow et al. (1994).

**Response:** Thank you for the clarification. We acknowledge that the accumulation (averaging) time and probe volume are the parameters controlling variance estimates, and that spending only a fraction of the time at a given height does not, by itself, introduce a bias in the variance, as discussed by Lenschow et al. (1994). The corresponding discussion has therefore

been removed from the manuscript.

**Comment 27:** l 453-457. I don't understand this discussion. Under stable conditions, the standard knowledge is that the length scale is smaller for stable conditions.

**Response:** Thank you for pointing this out ! We acknowledge that the original wording was incorrect. Under stable stratification, turbulence is indeed characterized by smaller length scales, whereas unstable conditions are associated with larger energetic eddies. We have corrected this mistake in the revised manuscript and reformulated the discussion accordingly to ensure consistency with standard boundary-layer turbulence theory. p.18, l.355-364.

**Comment 28:** l 465. How can systematic (that is stationary) spatial gradients introduce more variance in the traditional method? I would think that it only introduces a bias in the mean.

**Response:** We thank the reviewer for this important clarification. We agree that a purely stationary spatial gradient does not, by itself, introduce variance but only a bias in the mean. The additional variance arises because scanning lidars sample different spatial locations sequentially rather than simultaneously. In the presence of spatial gradients, sequential sampling combined with advection causes spatial variability to be mapped onto temporal fluctuations when the LOS measurements are combined, leading to apparent variance. We have clarified this point in the revised manuscript to avoid ambiguity. p.18, l.384-386.

**Comment 29:** l 473-476. Interesting that the error is significantly larger for the cross-wind component, but can you explain why? I cannot really follow the logic in the explanation. Maybe there is a hint in ?

**Response:** We have revised the discussion to clarify that the larger errors in the cross-wind component arise from the sequential nature of scanning lidar measurements, as discussed by Sathe and Mann (2012). Because LOS measurements are acquired at different times, the reconstruction of the cross-wind component relies on combining measurements that decorrelate more rapidly in time and space than those contributing to the along-wind component. Along-wind fluctuations are advected by the mean flow and therefore remain correlated over the scan cycle, whereas cross-wind fluctuations decorrelate more quickly. This leads to a reduced or distorted estimate of cross-wind variance, even when the mean wind direction is accurately known. This explanation has now been made explicit in the revised manuscript. p.19, l.394-405.

**Comment 30:** l 507. "identified the problematic". I guess you mean "problem", or "research issue".

**Response:** The term "problematic" has been replaced by "problem" in the revised manuscript. p.20, l.429.

Sincerely,
Maxime Thiébaut

**References**

Kelberlau, F., Neshaug, V., Lønseth, L., Bracchi, T., and Mann, J.: Taking the motion out of floating lidar: Turbulence intensity estimates with a continuous-wave wind lidar, Remote

Sensing, 12, 898, 2020.

Lenschow, D. H., Mann, J., and Kristensen, L.: How long is long enough when measuring fluxes and other turbulence statistics?, Journal of Atmospheric and Oceanic Technology, 11, 661–673, https://doi.org/10.1175/1520-0426(1994)011⟨0661:hlilew⟩2.0.co;2, 1994.

Manami, M., Mann, J., Sjöholm, M., Léa, G., and Gorju, G.: Squeezing turbulence statistics out of a pulsed Doppler lidar, Atmospheric Measurement Techniques, 18, 7513–7523, https://doi.org/10.5194/amt-18-7513-2025, 2025.

Mayor, S. D., Lenschow, D. H., Schwiesow, R. L., Mann, J., Frush, C. L., and Simon, M. K.: Validation of NCAR 10.6-micrometer $CO_2$ Doppler lidar radial velocity measurements and comparison with a 915-MHz profiler, Journal of Atmospheric and Oceanic Technology, 14, 1110–1126, https://doi.org/10.1175/1520-0426(1997)014⟨1110:vonmcd⟩2.0.co;2, 1997.

Sathe, A. and Mann, J.: Measurement of turbulence spectra using scanning pulsed wind lidars, Journal of Geophysical Research: Atmospheres, 117, 2012.

Sathe, A., Mann, J., Vasiljevic, N., and Lea, G.: A six-beam method to measure turbulence statistics using ground-based wind lidars, Atmospheric Measurement Techniques, 8, 729–740, https://doi.org/10.5194/amt-8-729-2015, 2015.

---

## Author Comment (AC2)

**Response to Reviewer 2**

Manuscript Title: *Dual-lidar profilers for measuring atmospheric turbulence*
Manuscript ID: wes-2025-179
Journal: *Wind Energy Science*

Dear Reviewer,

Thank you for the careful evaluation of our manuscript and for the constructive comments. The manuscript has been revised accordingly. Below we provide a detailed, point-by-point response to all comments. Your comments are shown in black, and our responses are shown in blue. All modifications to the manuscript are highlighted or tracked, in accordance with the journal's guidelines.

**Reviewer 2**

Thiébaut et al. present an interesting study on turbulence measurements with profiling lidar. The idea to place two lidars with a yaw angle offset and combining them to a lidar with more beams is interesting and innovative. The plots are well prepared and the manuscript is well written. At mutiple points I feel that the description of the results and the methods are a bit unprecise. The comparison to other lidar configurations and techniques on the contrary goes a bit too far in my opinion, because it cannot be justified with the results of the experiment. I thus suggest the manuscript for publication only after major revision. General and specific comments are given below.

**General comments**

- The database could be better described. A brief statement on wind speed span, median and mean is given, but for example no information about the distribution of wind direction. Is that dataset statistically significant? I think it is, but it is not shown.

- I am not very convinced about the comparison with the 6-beam method. You cannot easily compare the datasets and the lidar parameters are quite different. The comparison of errors and uncertainties on that basis is not sound.

- Some details of the results are not explained in enough detail. For example, the effects for cross-wind variance and how they depend on wind direction, or the differences in the spectra. Why is the low frequency not the same for all methods, why does the variance method have more high frequency noise etc.

**Response:** We thank the reviewer for these constructive general comments and address them as follows.

(i) Description and statistical representativeness of the database:

We agree that the original manuscript did not sufficiently document the characteristics of the dataset. The revised manuscript now provides additional information on the dataset size, wind speed range, and the distribution across atmospheric stability classes. While wind direction statistics are not explicitly shown, the dataset comprises 1,098 independent 30-min periods spanning a wide range of meteorological conditions, which we consider statistically significant for the purposes of this study. This point has been clarified in the revised text. Sect. 2.1 (p.4)

and Fig. 2 (p.5).

(ii) Comparison with the six-beam method:
We agree that a quantitative comparison between the variance method and the six-beam method is not robust given the differences in datasets, scanning strategies, and lidar parameters. This concern was also raised by the first reviewer. Consequently, the comparison of errors and uncertainties with the six-beam method (including Table 2 and Fig. 5) and the associated discussion have been removed from the revised manuscript.

(iii) Interpretation of specific result features:
We acknowledge that some interpretations—particularly regarding cross-wind variance behavior, wind-direction sensitivity, and spectral differences—were not sufficiently supported by the results presented. As these dependencies were not explicitly demonstrated and could not be robustly quantified within the scope of this study, the corresponding interpretations have been removed or substantially revised. The revised manuscript now focuses on results that are directly supported by the data and on mechanisms that can be clearly attributed to probe-time and spatial averaging effects. Regarding the spectra, the apparent differences between the methods do not indicate physical discrepancies or increased noise. In particular, the white-noise plateau is not visible in the traditional method because the high-frequency portion of the spectra is contaminated by inter-beam effects, which distort the spectral shape and mask the underlying noise behavior. This clarification has now been added to the manuscript. p.13, l.290-292.

**Specific comments**

    **Comment 1:** p.1, l.21. I do not think that you can say that so generally. There are a lot of people who do VAD with pulsed lidars as well. It has advantages, especially for turbulence retrievals, too.

    **Response:** Thank you for the clarification. We agree that the distinction between pulsed and continuous-wave lidars in terms of DBS and VAD operation is not exclusive, and that VAD scanning strategies are also commonly applied to pulsed lidar systems, particularly for turbulence retrievals. We have therefore revised the text to avoid this overly general statement and now emphasize that DBS and VAD represent commonly used, but not exclusive, measurement strategies for pulsed and continuous-wave lidars, respectively. p.1, l.21-24.

    **Comment 2:** p.2, l.48. Eberhard et al. (1989) requires a full VAD at 35.3° and provides TKE and the covariances, not the single component variances. Later studies by Smalikho, Stephan, Wildmann and Päschke showed that this method is very accurate, if the lidar "intra-beam" volume averaging effects are corrected in the retrieval.

    **Response:** Thank you for this important clarification. We agree that Eberhard et al. (1989) is based on a full VAD scan at 35.3° elevation and retrieves turbulence kinetic energy and velocity covariances, rather than individual component variances. Our original wording was therefore imprecise. We have revised the text to correctly describe the scope of the Eberhard et al. (1989) approach and to distinguish it from the variance method applied here. We also now acknowledge subsequent studies demonstrating the high accuracy of VAD-based turbulence retrievals when intra-beam averaging effects are properly accounted for. The manuscript has

been corrected accordingly. p.2, l.42-46.

**Comment 3:** p.3, l.63. There has recently been a release of a commercial 6-beam lidar https://halo-photonics.com/lidar-systems/beam-6x/, https://halo-photonics.com/lidar-systems/beam-6x-windpower/. It also does not take 15s for an instantaneous measurement any more. I saw further down the manuscript that you discuss this instrument, but i think it would be fair to mention here already.

**Response:** We have revised the manuscript to explicitly acknowledge the recent commercialization of six-beam lidar profilers inspired by the WindScanner concept, in particular the Beam6X WindPower developed by Lumibird. This is now mentioned at the point where the dual-lidar methodology is introduced, and we clarify that the dual-WindCube configuration used in this study provides the minimum number of independent beams required by the variance method while relying on well-established lidar profilers that are already widely used and trusted in industrial applications. This addition places our approach in the context of emerging six-beam technologies while motivating the experimental choices made in the present study. p.2, l.54-56.

**Comment 4:** p.4, l.112. Despite the fact of the foundation being quite impressive, I am not sure how relevant it is for this study. Information about the wind conditions at the site (wind rose, etc.) could be quite interesting instead.

**Response:** We agree that the detailed description of the mast foundation and wind farm infrastructure was not directly relevant to the objectives of this study. This information has therefore been removed from the manuscript. In response to the second part of the comment, we have added a characterization of the mean wind conditions at the site, including a wind rose and a wind speed distribution for the analysis period, together with a concise description of the dominant wind directions and wind speed statistics. This revision provides more relevant contextual information for the interpretation of the lidar measurements. p.4, l.99-107 and Fig. 2.

**Comment 5:** p.7, l.159. Also for the collected dataset, some statistics would be helpful here: wind rose, histograms, of wind, turbulence, stability for example.

**Response:** We have added a characterization of the mean wind conditions for the collected dataset, including a wind rose and a wind speed histogram for the analysis period. These statistics are now presented in a new figure and accompanying text describing the dominant wind directions and wind speed distribution. Turbulence and stability statistics are treated separately in later sections of the manuscript, where they are directly relevant to the evaluation of the turbulence retrieval methods. p.4, l.103-107, Fig. 2.

**Comment 6:** p.8, l.185. Please provide the thresholds for the despiking.

**Response:** We thank the reviewer for this comment. The manuscript has been revised to explicitly state the despiking thresholds. Following Wang et al. (2015), spikes are identified within consecutive 30-min windows when the absolute differences between adjacent velocity samples exceed twice the interquartile range (2×IQR) and exhibit opposite signs. This clarification has been added to the text. p.8, l.162-167.

**Comment 7:** p.10, Eq. 13-14. $U$ remains the absolute velocity?

**Response:** Yes, $U$ denotes the wind speed magnitude, defined as the modulus of the horizontal wind velocity vector. p.9, l.178.

**Comment 8:** p.11, Eq.19. You switch here from vector notation to Einstein notation (I think), without explaining it. That could be confusing for readers and should be explained.

**Response:** Thank you for pointing this out. The original formulation could indeed be interpreted as relying on implicit index summation. We have revised the manuscript to remove this ambiguity by explicitly defining the construction of the matrix **Q** from the transformation matrix **T** and by clarifying that no implicit summation over repeated indices is assumed. In addition, we have added a numbered equation that explicitly writes the relationship between the LOS variances and the Reynolds stress components in index form. These changes clarify the notation and ensure consistency between the matrix and index formulations. p.10, Eq. 7.

**Comment 9:** p.13, l.310. I assume that the "virtual kinematic heat flux" was calculated using the sonic vertical velocity and sonic temperature? Thus not directly the virtual temperature. Could be confusing if you use the same symbol as for the average virtual temperature from the WXT530.

**Response:** Thank you for highlighting this ambiguity. The kinematic heat flux used in the computation of the Monin–Obukhov length is derived from high-frequency sonic anemometer measurements and is therefore based on sonic temperature fluctuations rather than virtual temperature fluctuations. To reflect this more accurately and avoid confusion with the mean virtual potential temperature derived from the WXT530, we have revised Eq. 36, which is now Eq. 19, and the associated text to use the covariance $\sigma_{w\theta_s}$. The mean virtual potential temperature $\theta_v$ is retained in the numerator and is computed independently from WXT530 temperature and humidity measurements. p.12, l.262-264 and Eq. 19.

**Comment 10:** p.15, Tab.2 and p.16, Fig.5. Comparing the methods with completely different datasets is not sound. You would have to make sure that you have the same amount of data for all sorts of wind bins, wind sectors, stability classes, which I assume is not the case here!?

**Response:** We agree that comparing the methods using datasets with different sampling distributions across wind speed bins, wind sectors, and stability classes is not statistically sound. This issue was also raised by the first reviewer. In response, we have removed the comparison presented in Table 2 and Fig. 5, along with the associated discussion, from the revised manuscript.

**Comment 11:** p.16, l.355f. Can you explain why the spectra differ at low frequencies?

**Response:** This point was already addressed in response to the first reviewer. The differences observed at low frequencies are attributable to differences in sampling strategy and effective averaging between the measurement approaches, which affect the representation of large-scale, low-frequency motions. We have clarified this explanation in the manuscript to make the origin of the low-frequency discrepancies more explicit. p.18, l.365-373.

**Comment 12:** p.22, ll.434ff. I think the ideas and comparison to other lidar configurations are a bit superficial and not exactly based on results from this study. I recommend to skip them and focus more on the direct findings of the new variance method.

**Response:** We agree with the reviewer that the previous discussion of specific lidar configurations was speculative and not directly supported by the results of this study. The section has been revised to remove device-specific comparisons and now focuses on probe-time averaging effects and accumulation time, which are directly supported by the findings and discussed in the context of Thiébaut et al. (2025) and Manami et al. (2025), as requested by the first reviewer. p.17, l.347-354.

**Comment 13:** p.22, l.457. Intermittent turbulence is especially observed in stable boundary layers with strong shear. A violation of homogeneity assumptions cannot be directly associated with neutral and unstable conditions alone. Neutral conditions can be perfectly homogeneous over flat terrain, stable conditions can be non-homogeneous with only slightly complex terrain. I think you should be a bit more precise what you mean here.

**Response:** We have revised the text to avoid associating intermittency or violations of homogeneity exclusively with neutral and unstable conditions. The discussion now clarifies that such features can also occur in stable boundary layers, particularly under strong shear, while emphasizing that higher turbulence levels and a broader range of energetic scales typically observed under neutral and unstable stratification tend to amplify inter-beam effects in the traditional reconstruction. p.18, l.360-364.

**Comment 14:** p.23, l.477. I think this dependency on wind direction should be shown explicitly.

**Response:** We agree that the dependence of the cross-wind turbulence intensity on wind-direction uncertainty should be demonstrated explicitly in order to support the interpretation. As this dependency is not directly quantified or shown in the present study, and the explanation was also not convincing to the first reviewer, we have removed this paragraph and the associated interpretation from the revised manuscript.

**Comment 15:** p.24, l.497. You should at least mention that a two-lidar setup doubles the cost at this point.

**Response:** We have added a sentence in the conclusion acknowledging that the dual-lidar configuration entails increased instrumentation costs due to the use of two lidar systems. p.19, l.413-414.

Sincerely,
Maxime Thiébaut

**References**

Eberhard, W. L., Cupp, R. E., and Healy, K. R.: Doppler lidar measurement of profiles of turbulence and momentum flux, Journal of Atmospheric and Oceanic Technology, 6, 809–819, https://doi.org/10.1175/1520-0426(1989)006⟨0809:dlmopo⟩2.0.co;2, 1989.

Manami, M., Mann, J., Sjöholm, M., Léa, G., and Gorju, G.: Squeezing turbulence statistics out of a pulsed Doppler lidar, Atmospheric Measurement Techniques, 18, 7513–7523, https://doi.org/10.5194/amt-18-7513-2025, 2025.

Thiébaut, M., Marié, L., Delbos, F., and Guinot, F.: Evaluating the enhanced sampling rate for turbulence measurement with a wind lidar profiler, Wind Energy Science, 10, 1869–1885, https://doi.org/10.5194/wes-10-1869-2025, 2025.

Wang, H., Barthelmie, R. J., Clifton, A., and Pryor, S. C.: Wind measurements from arc scans with Doppler wind lidar, Journal of Atmospheric and Oceanic Technology, 32, 2024–2040, https://doi.org/10.1175/jtech-d-14-00059.1, 2015.